# Basic Substances and Potential Basic Substances: Key Compounds for a Sustainable Management of Seedborne Pathogens

Laura Orzali [1,*], Mohamed Bechir Allagui [2], Clemencia Chaves-Lopez [3], Junior Bernardo Molina-Hernandez [3], Marwa Moumni [4], Monica Mezzalama [5] and Gianfranco Romanazzi [4,*]

1   Council for Agricultural Research and Economics (CREA), Research Center for Plant Protection and Certification (CREA-DC), Via C.G. Bertero 22, 00156 Rome, Italy
2   Plant Protection Laboratory, National Institute of Agricultural Research of Tunisia (INRAT), Carthage University, Rue Hedi Karray, Ariana 2080, Tunisia; allagui.bechir@gmail.com
3   Bioscience and Agro-Food and Environmental Technology Department, University of Teramo, Campus "Coste Sant'Agostino", Via R. Balzarini 1, 64100 Teramo, Italy; cchaveslopez@unite.it (C.C.-L.); jbmolinahernandez@unite.it (J.B.M.-H.)
4   Department of Agricultural, Food and Environmental Sciences, Marche Polytechnic University, Via Brecce Bianche, 60131 Ancona, Italy; m.moumni@staff.univpm.it
5   AGROINNOVA—Interdepartmental Centre for Innovation in the Agricultural and Food Sector, University of Torino, Largo Paolo Braccini 2, 10095 Grugliasco, TO, Italy; monica.mezzalama@unito.it
*   Correspondence: laura.orzali@crea.gov.it (L.O.); g.romanazzi@univpm.it (G.R.)

**Abstract:** Seedborne pathogens represent a critical issue for successful agricultural production worldwide. Seed treatment with plant protection products constitutes one of the first options useful for reducing seed infection or contamination and preventing disease spread. Basic substances are active, non-toxic substances already approved and sold in the EU for other purposes, e.g., as foodstuff or cosmetics, but they can also have a significant role in plant protection as ecofriendly, safe, and ecological alternatives to synthetic pesticides. Basic substances are regulated in the EU according to criteria presented in Article 23 of Regulation (EC) No 1107/2009. Twenty-four basic substances are currently approved in the EU and some of them such as chitosan, chitosan hydrochloride, vinegar, mustard seed powder, and hydrogen peroxide have been investigated as seed treatment products due to their proven activity against fungal, bacterial, and viral seedborne pathogens. Another basic substance, sodium hypochlorite, is under evaluation and may be approved soon for seed decontamination. Potential basic substances such as essential oils, plant extracts, and ozone were currently found effective as a seed treatment for disease management, although they are not yet approved as basic substances. The aim of this review, run within the Euphresco BasicS project, is to collect the recent information on the applications of basic substances and potential basic substances for seed treatment and describe the latest advanced research to find the best application methods for seed coating and make this large amount of published research results more manageable for consultation and use.

**Keywords:** chitosan; essential oils; phytotoxicity; seed coating; seed quality; seed treatment; sustainability

## 1. Introduction

The seed is an essential input for crop production, since 90% of food crops are grown from seeds. For this reason, the use of healthy seeds is an essential key to successful agricultural production and serves as the backbone for good economic harvest. Seeds can carry a heavy load of microorganisms, which can cause severe diseases and be responsible for various negative effects on yield and the spread of pathogen inoculum in the soil. Seed movement is also the main cause of pathogenic spread across international borders and the

introduction of diseases into previously unaffected areas or their re-emergence [1]. There are many examples of seedborne pathogens that have spread globally, some of which can cause devastating diseases in some of the most important staple crops. Just as examples, Karnal bunt of wheat, caused by *Tilletia indica*, was introduced from India to Mexico in 1972, from Mexico to the USA in 1996 [2], and from an unknown source to South Africa in 2002 [3]; wheat blast caused by the *Magnaporthe oryzae Triticum* pathotype was spread from South America to Bangladesh [4] and to Zambia [5]; wheat streak mosaic virus was spread from Mexico to Australia [6]; and maize lethal necrosis caused by maize mottle chlorotic virus was spread from Asia to Kenya [7].

Seed treatments represent the first line of defense against seedborne (surface-borne or internally seedborne) and soilborne pests. Seed treatments are defined as "the biological, physical, and chemical agents and techniques applied to seed to provide protection and improve the establishment of healthy crops" [8], and in the last 200 years from the discovery of the Bordeaux mixture, several active ingredients have been developed to be used as coating to protect seeds and seedlings in the early stages of their growth. Munkvold [9] exhaustively reviewed the history and development of chemical control of seedborne pathogens. Over the past decade, the number of studies on seed treatment has increased significantly, reflecting the growing interest of the scientific community [10]. Lamichhane et al. [11] summarized the potential negative effects of synthetic fungicides used for seed treatments on nontarget organisms. These effects could consist of a reduction in biocontrol agents and earthworms' activities, alteration of litter decomposition rate, decline in the number of rhizobia on seeds and in the arbuscular mycorrhiza colonization, as well as a reduction in fungal endophytes of seedlings. These fungicide-induced disturbances also had negative consequences on root and shoot biomass and grain yield. Following European Community initiatives, many lines of research and scientific efforts have focused on the development of environmentally friendly alternatives to the use of pesticides for managing crop diseases, in particular seedborne diseases [12–14].

Reducing the use of synthetic pesticides is a major challenge in many countries, and the search for alternative crop protection products is a strategy for promoting more sustainable agricultural systems. Nowadays, the use of traditional environmentally friendly practices (e.g., sanitation, crop rotation, adjusting the age of planting) to control diseases is integrated with new advanced techniques or tools to avoid or at least limit the use of synthetic pesticides. Several sustainable seed treatments can be used including physical treatments such as heat treatments, with the most common being hot water, hot air, and electron treatments, biocontrol agents with species belonging to the genus *Trichoderma*, or plant growth-promoting rhizobacteria (PGPR) and the use of natural substances with antimicrobial activity and/or priming effects [10]. Alternative methods such as seed treatment using basic substances or potential basic substances to manage seedborne pathogens can be a solution to ensuring safe agricultural production, but these substances are still poorly known by researchers and growers and have not been placed on the market as plant protection products [15]. Basic substances are relatively novel compounds already approved and sold in the EU for other purposes, e.g., as foodstuff or cosmetics, which can be used in plant protection without neurotoxic or immune-toxic effects as ecofriendly, safe, and ecological alternatives to synthetic pesticides [16,17]. Among the 24 basic substances approved in the EU, five of them were approved as a seed treatment: chitosan hydrochloride, chitosan, vinegar, mustard seed powder, and hydrogen peroxide. Moreover, potential basic substances such as ozone, essential oils, and plant extracts have been used as seed treatment.

The number of studies on seed treatment with such natural/ecofriendly substances has increased over the last decade resulting in a large amount of published investigations. The aim of this review, carried out in the framework of the Euphresco BasicS project, is to provide an overview of the use of already approved basic substances and of potential basic substances as seed treatments for the control of seedborne pathogens, in order to make this large amount of published results more manageable for consultation and use. Since

there are many different techniques that can be used for this purpose, the latest advanced research in finding the best application method as seed coating, dressing, or spraying is also described.

## 2. Methods for Seed Treatment

### 2.1. Seed Immersion

Seed immersion methods are those in which seeds are soaked in aqueous or solvent-based liquid for a certain length of time, depending on the nature of the seed coat and the substance used. The soaking results in partial or full hydration of both the host and pathogen and produces microscopic ruptures, making them more susceptible to the penetration of active substances compared to the dry state [12]. Not all the substances are soluble in water, so in some cases (e.g., essential oil, chitosan), it is necessary to use an emulsifier to allow for mixing and emulsion homogeneity [18]. Besides the direct antimicrobial effects that depend on the type of substance used, immersion treatment can have the following priming effects: increased germination rate and seedling vigor; induced diverse range of morphophysiological, biochemical, and molecular responses in plants; and thus improved abiotic and biotic stress tolerance and increased crop yields [19]. Immersion represents the most widely used method for treatment with elicitors for resistance induction, such as chitosan and methyl salicylate [20,21]. Timing of the treatment plays a key role in phytotoxicity, negatively influencing seed vitality [20]. Moreover, excessive imbibition during seed submersion can damage the outer seed coats, especially in the case of seeds with softer teguments such as legume seeds [22]. The challenge is to find the right combination of treatment durations for different seed types to ensure efficacy without causing phytotoxicity. Primed seeds are known to have low storage longevity, which can be partially remedied via post-storage treatments such as dehydration, heat shock, or post-storage humidification [23]. The soaking process is considered cumbersome and time-consuming when treating large quantities of seeds at a large scale, because it requires a large volume of liquid and needs subsequent drying [18].

### 2.2. Seed Dressing and Coating

Innovative seed coating and dressing technologies are useful as delivery systems for the application of active ingredients on the seed surface. The technique of seed dressing involves the application on the seed surface of a thin layer of the active product, such as pesticides, fertilizers, or growth promoters which can be applied both as dry or liquid formulations [12]. Seed dressing is the most widely used method for low dosages of active components onto seeds [24] and although there are many types of equipment used for coating, the most commonly used device is performed with a rotary coater [18]. Seed coating is a technique in which an external material is applied to the surface of the seed using a binder which acts as an adhesive to improve the adhesion of the active ingredients to the seed. The role of the binder is also to ensure coating integrity during and after drying and to prevent cracking and dusting off during handling and sowing [18]. The layer is applied to the seed typically from 2 to 5% of the seed weight [25]. In this context, nanotechnology could represent an innovative tool exploitable in agriculture, since nanoparticles (materials with a size ranging from 1 to 100 nm) [26,27] can be effective carriers of seed health-promoting compounds when applied as seed coatings or seed dressing material [27]. Nanoagroproducts are an upcoming technology that might be beneficial for the development of future generations of formulations for seed treatment to enhance the sustainability of agricultural systems. Among them, a wide selection of organic and natural compounds can be loaded into these nanoparticles, including essential oils, cellulose, and chitosan, making this technology suitable for sustainable and ecofriendly farming. Basic substances can take advantage of this technology to take place in adapted formulations of seed coating products. Seed coating allows for a controlled release of the substance reducing the active ingredient dosage needed, thus reducing their release into the ecosystem and soils, the possible toxicity for plants and the environment, and the treatment cost. Nanoscale

materials used in seed coating technologies such as nanocapsules, nanogels, nanofibers, nanoclays, and nanosuspensions are supposed to increase the accuracy and efficiency of seed protection products, allowing for a reduction in pesticides in the field [27]. On the other hand, specific machines and equipment are required for seed dressing and coating techniques which are performed with a dry power applicator, rotary or drum machine, motor, or hand driving [18].

## 3. Seed Treatment with Approved Basic Substances

### 3.1. Activity of Approved Basic Substances against Fungi and Oomycetes

Chitosan is a naturally occurring biopolymer with antimicrobial properties explored in agriculture for many uses as a plant defense inducer, growth promoter, and carrier for delivery systems of biocontrol agents [28]. In 2014, chitosan hydrochloride was approved by the EU as one of the first basic substances for plant protection [29], and a second chitosan formulation was approved in 2022 [30]. Chitosan has shown activity against several species of seedborne pathogens (Tables 1 and 2). El-Mohamedy et al. [31] reported that soaking seeds of green bean (*Phaseolus vulgaris*) in chitosan (1 g L$^{-1}$) reduced the pre-emergence incidence of *Rhizoctonia solani* and *Fusarium solani* by 54.4% and 52.6%, respectively, after 40 days of plant growth in a greenhouse in soils naturally infested with either of these fungi. No sign of phytotoxicity was reported on the plants obtained by germinated seeds. Fenugreek (*Trigonella foenum-graecum*) seeds were treated with different concentrations of chitosan and then inoculated 24 h later with *F. solani* conidia. Results showed that six days post-inoculation, root rot disease incidence was reduced by 87.5% and 90.1%, with no significant difference for the seeds treated with chitosan (2 g L$^{-1}$) or carbendazim (0.5 g L$^{-1}$), respectively [32]. In this experiment, the radicle length of fenugreek seedlings due to chitosan (0.5 g L$^{-1}$) was significantly higher (3.76 cm) over the control (2.26 cm) and carbendazim (3.34 cm). Bhardwaj et al. [33] evaluated different pearl millet (*Pennisteum glaucum*) seed treatments including chitosan that were sown in several experiment fields in India regarding blast disease caused by *Pyricularia grisea*. The application rate of chitosan seed immersion was 0.5 g kg$^{-1}$ per liter of water and resulted in a blast severity reduction ranging from 4.7% to 26.9%, depending on the field location and growing season. Spelt (*Triticum spelta*) seeds immersed in a conjugate complex solution of chitosan (1.5 g L$^{-1}$) and tyrosine (15 g L$^{-1}$), then inoculated with a conidia suspension of *Fusarium culmorum*, showed a 50% reduction in the incidence of root rot in the seedlings [34]. No phytotoxicity was observed. Seeds of groundnut (*Arachis hypogaea*) were coated with chitosan polymer (1 g L$^{-1}$), sowed in potted soil infested with *Aspergillus niger* and grown for 50 days under greenhouse conditions. Incidence of *Aspergillus* collar rot on seedlings from coated seeds was reduced by 51.8% compared to inoculated untreated seeds [28]. Similar studies were carried out by the same authors on safflower (*Carthamus tinctorius*) seeds coated and sown in infested soil with *Macrophomina phaseolina*. Chitosan coating did not affect the germination rate of either the groundnut or safflower seeds. Reduction of the pathogen was 15.7% on seedlings from seeds treated with chitosan. Chitosan was also tested on cucumber (*Cucumis sativus*) seeds against the oomycete *Phytophthora capsici*. Cucumber seedlings coming from the seeds immersed in chitosan at 500 ppm (0.05%) were grown in plastic pots in a screenhouse; chitosan treatment provided 85% disease suppression of damping off caused by seedling inoculation with zoospores of *P. capsici* injected into the rhizosphere [35]. Moreover, seed germination and root and shoot growth of cucumber were enhanced by chitosan seed treatment in a dose-dependent way up to 500 ppm. Chitosan (0.5%, *w/v*) seed immersion treatment was also used to reduce foot and root rot caused by *Fusarium graminearum* in durum wheat (*Triticum durum*) plants from both naturally and artificially infected seeds. This treatment caused the stimulation of a defense system as phenolic content increasing and defense-related enzyme activation in seedlings. In the field, seedlings from natural and artificial seed infection showed a reduction of foot and root rot disease by 36% and 56%, respectively. In the greenhouse, the disease reduction was 38% for seedlings from seeds that were artificially infected [20].

**Table 1.** In vivo and in-field activities of basic substances applied as seed treatments to control seedborne fungi and oomycetes in different crops. Effectiveness is reported as the disease incidence or symptom percentage reduction compared to the untreated control. Phytotoxicity was evaluated through germination testing and the results are compared to the untreated control.

| Crop | Disease/Pathogen | Substance (Concentration) | Application | Effectiveness | Possible Phytotoxicity | Activity/Defense Response | Reference |
|---|---|---|---|---|---|---|---|
| Green bean (*Phaseolus vulgaris*) | *Rhizoctonia solani* [1,*] | Chitosan (1 g L$^{-1}$) | Immersion | 54.4% | Data not available | | [31] |
| | *Fusarium solani* [1,*] | Chitosan (1 g L$^{-1}$) | Immersion | 52.6% | Rate of seed germination equal to the control | | |
| Fenugreek (*Trigonella foenum-graecum*) | *Fusarium solani* [1,*] | Chitosan (2 g L$^{-1}$) | Immersion | 87.5% | Rate of seed germination equal to the control | Radicle length improvement | [32] |
| Pearl millet (*Pennisteum glaucum*) | *Magnaporthe grisea* [2] | Chitosan (0.5 g L$^{-1}$) | Immersion | 4.7%–26.9% | Data not available | | [33] |
| Spelt (*Triticum spelta*) | *F. culmorum* [2] | Chitosan (1.5 g L$^{-1}$) | Immersion | 50.0% | Rate of seed germination equal to the control | Seed germination increasing | [34] |
| Groundnut (*Arachis hypogaea*) | *Aspergillus niger* [2,*] | Chitosan (1 g L$^{-1}$) + *Trichoderma* spores | Immersion | 51.8% | Rate of seed germination equal to the control | | [28] |
| Safflower (*Carthamus tinctorius*) | *Macrophomina phaseolina* [2,*] | | Immersion | 15.7% | | | |
| Cucumber (*Cucumis sativus*) | *Phytophthora capsici* [1,*] | Chitosan (500 ppm) | Immersion | 85.0% | Increased seed germination | Seedling shoot and root growth increasing | [35] |
| Durum wheat (*Triticum durum*) | Fusarium foot rot *F. graminearum* [1,2] | Chitosan (0.5% *v/v*) | Immersion | In field [1]: 36% In field [2]: 56% In greenhouse [2]: 38% | Rate of seed germination equal to the control | Phenolic content increasing and defense-related enzyme activation | [20] |
| Common wheat (*Triticum aestivum*) | *F. culmorum* [2] | White mustard meal (15 g mustard + 45 mL H$_2$O per kg) | Wet and dry seed dressing | In vitro: 67% In field: 43%–78% | Rate of seed germination equal to the control | Plant development stimulation: improving grain quality and wheat plant growth | [36] |
| Pine (*Pinus radiata*) | *F. circinatum* [2] | Hydrogen peroxide (33% *w/v*) | Immersion | 98.2% | Seedling emergence reduction | | [37] |
| Carrot (*Daucus carota*) | *Alternaria radicina* [1] | Hydrogen peroxide stabilized with silver ions (0.025%) | Immersion | 43.2% | Rate of seed germination equal to the control | | [38] |
| White lupin (*Lupinus albus*) | *Colletotrichum lupini* [1] | Vinegar (5% acetic acid) | Immersion for 30 min | 16.9% | Rate of seed germination equal to the control | | [39] |

Table header: **Basic Substances—Fungi and Oomycetes**

[1] Natural contamination; [2] artificial inoculation; * soil contamination.

Besides chitosan, other compounds like mustard seed power, vinegar, and hydrogen peroxide were approved as basic substances by the European Union between 2015 and 2017 [15] and allowed for agricultural uses (Table 1). Kowalska et al. [36] recommended the dose of 15 g mustard meal per 1 kg common wheat grain (*Triticum aestivum* ssp. *vulgare*) as a seed dressing applied with 45 mL of water, to significantly reduce disease caused by *F. culmorum* on wheat during the early stage of growth. The authors reported a stimulating effect of mustard meal seed dressing on seedling development without perceiving any negative influence on the germination and development of seedlings, accompanied by a reduction in the number of infected seeds and by a 43–78% disease incidence reduction in the field, according to the type of seed dressing applied, respectively, wet or dry. Berbegal et al. [37] evaluated *Pinus radiata* seed treatments using hydrogen peroxide (33% *w/v*,

disinfectant conc. 30%) to control *Fusarium circinatum*. Seeds artificially inoculated and treated by soaking in hydrogen peroxide were sown in peat moss and then maintained in a forest nursery. The reduction of disease incidence in seedlings from seeds treated with hydrogen peroxide ranged from 98.2% to 100% but the germination rate was also reduced compared to inoculated untreated seeds. Differently, hydrogen peroxide stabilized with silver ions applied to *Daucus carota* seeds had no phytotoxic effects, and it caused a significant decrease in the percentage of seeds infested with *Alternaria radicina* [38].

**Table 2.** In vivo and in-field activities of basic substances applied as seed treatments to control seedborne bacteria in different crops. Effectiveness is reported as the disease incidence or symptom percentage reduction compared to the untreated control. Phytotoxicity was evaluated through germination testing and the results are compared to the untreated control.

| | | | Basic Substances—Bacteria | | | | |
|---|---|---|---|---|---|---|---|
| Crop | Disease/Pathogen | Substance (Concentration) | Application | Effectiveness (Disease/Symptoms Reduction) | Possible Phytotoxicity | Activity/Defense Response | Reference |
| Lettuce (*Lactuva sativa*) | *Xanthomonas campestris* pv. *vitians* [2] | Hydrogen peroxide (3% *w/v*) ———————— Hydrogen peroxide (5% *w/v*) | Immersion | 100% | Rate of seed germination equal to the control ———————— Significant reductions in germination | Direct antibacterial activity | [40] |
| Cabbage (*Brassica oleracea*) | *Xanthomonas campestris* pv. *campestris* [1] | Hydrogen peroxide (10%; 20% *w/v*) | Immersion | Depending on the concentration up to 100% | Rate of seed germination equal to the control | Direct antibacterial activity | [41] |

[1] Natural contamination; [2] artificial inoculation.

Table vinegar (pH = 3, acetic acid 5%) was also tested in order to reduce *Colletotrichum lupini* seed infection on lupin (*Lupinus albus*) [39]. Anthracnose-infected seeds from highly infected plots were soaked in vinegar and grown under field conditions. The authors reported that vinegar treatment successfully reduced disease severity (16.9%) and increased yield to levels similar to those observed for certified seeds, without significantly affecting germination rate [39].

*3.2. Activity of Approved Basic Substances against Bacteria*

The bactericidal action of oxygen released from peroxides is well known, and the possibility of direct horticultural benefits plus bactericidal activity make hydrogen peroxide attractive in agriculture for seed disinfection. However, there are only a few recent reports on in vivo or field applications (Table 2). Since 2002, hydrogen peroxide was investigated as a seed treatment for the control of bacterial leaf spot of lettuce (*Lactuca sativa*) caused by the seedborne bacterium *Xanthomonas campestris* pv. *vitians.* Bacteria were not detected when seeds were treated with 3 or 5% hydrogen peroxide, even if the treatments at 5% concentration reduced seed germination up to 28% compared with controls [40]. More recent works about seed treatment with hydrogen peroxide against bacterial diseases have only come after years of research: hydrogen peroxide at 3% was investigated as a seed treatment against *Xanthomonas campestris* pv. *campestris* in cabbage (*Brassica oleracea*) seeds [41]. The treatment for 30 min was the most effective, both in terms of disinfection rate and of seed viability, but the side effects on the seed coat observed when the procedure was carried out at the company facilities suggested 15 min as the maximum time of immersion without losing effectiveness.

**4. Seed Treatment with Potential Basic Substances against Pathogens**

*4.1. Activity of Potential Basic Substances against Fungi and Oomycetes*

Essential oils (EOs) are secondary metabolites accumulated by aromatics or medical plants and extracted from leaves, flowers, roots, and barks. They exhibit antifungal activity due to the presence of different bioactive ingredients (alkaloids, phenols, monoterpenes

and sesquiterpenes, isoprenoids) in different concentrations, their composition may vary even within the same species, affecting antimicrobial activity [42,43]. EOs have widely demonstrated over the years their efficacy against various fungal pathogens in vitro [44,45] and in recent years, the scientific research in this field has focused primarily on in vivo and field applications (Tables 3 and 4). Immersion seed treatment with clove (*Syzygium aromaticum*) EO was able to reduce *Fusarium* spp. infection on maize and wheat seeds at different doses, but the effective rates ($5 \times 10^3$ and $5 \times 10^4$ ppm, respectively, for maize and wheat) had a high phytotoxicity effect [46]. Clove oil has also been tested in field trials, both as a seed soak and as coating (spray) on wheat and field peas against, respectively, *Tilletia laevis* [47] and *Ascochyta blight* complex [22], artificially inoculated on seeds, with good effectiveness, which varied from year to year. Submersion application has demonstrated a more reliable effectiveness over the years, compared to coating application. In the tomato, eucalyptus (*Eucalyptus grandis*), caraway (*Cuminum cyminum*), and citrus (*Citrus sinensis*), EOs have been tested as seed treatments against *Fusarium oxysporum* [48], and oregano EO (*Origanum vulgare*), against *F. oxysporum* f.sp. *lycopersici* [49] artificially inoculated in soil, with a reduction in disease incidence and severity. Tomato seedlings showed no phytotoxic effects after soaking treatment at the applied rates (Table 3).

**Table 3.** In vivo and in- field activities of potential basic substances applied as seed treatments to control seedborne fungi and oomycetes in different crops. Effectiveness is reported as the disease incidence or symptoms percentage reduction compared to the untreated control. Phytotoxicity was evaluated through germination testing and the results are compared to the untreated control.

| Potential Basic Substances—Fungi and Oomycetes | | | | | | | |
|---|---|---|---|---|---|---|---|
| Crop | Target Disease/Pathogen | Substance (Concentration) | Application | Effectiveness (Disease/Symptoms Reduction) | Possible Phytotoxicity | Activity/Defense Response | Reference |
| Durum wheat (*Triticum durum*) | Common bunt/*Tilletia laevis* * | *Syzygium aromaticum* EO (0.3% *v/v*) | Immersion for 10 min | From 30% to 90% | Seed germination reduction | Reduction in pathogen incidence | [47] |
| | | *S. aromaticum* formulation (2.5% *v/v*) | | From 40% to 100% | Rate of seed germination equal to the control | | |
| | | *S. aromaticum* EO (1% *v/v*) | Coating | From 30% to 82% | Rate of seed germination equal to the control | | |
| | | *S. aromaticum* formulation (5% *v/v*) | Coating | From 30% to 85% | | | |
| Wheat (*Triticum aestivum*) | *Fusarium equiseti* [2]; *F. culmorum* [2]; *F. poae* [2]; *F. avenaceum* [2] | *S. aromaticum* EO $5 \times 10^3$ ppm | Immersion for 8 min | 100% | Total inhibition of seed germination | Inhibition of pathogen development | [46] |
| | *Alternaria* spp. *Fusarium* spp. *Drechslera* spp. | *Origanum vulgare*, *Thymus vulgaris* and *Coriandrum sativum* Eos | Vapour | 50% | Inhibition of seed germination at 0.4% (thyme and oregano EO) | Inhibition of deoxynivalenol (DON) occurrence | [50] |
| | *Aspergillus* spp. *Fusarium* spp. [1,2] | Ozone (60 mg L$^{-1}$) | Ozonation for 300 min | 54.3% | – | | [51] |

**Table 3.** *Cont.*

| | | | Potential Basic Substances—Fungi and Oomycetes | | | | |
|---|---|---|---|---|---|---|---|
| Crop | Target Disease/Pathogen | Substance (Concentration) | Application | Effectiveness (Disease/Symptoms Reduction) | Possible Phytotoxicity | Activity/Defense Response | Reference |
| Pea *Pisum sativum* | Ascochyta blight fungal complex (*Dydimella pinodes*, *D. pinodella*, *D. pisi*) [2] | *S. aromaticum*-based formulation (0.2% *v/v*) | Immersion for 10 and 20 min | From 68% to 71% | Rate of seed germination equal to the control but in field an excessive handling after imbibition could damage seeds | In vivo: reduction in seed infection percentage In field: seedling protection and established plants enhancement | [22] |
| | | *Thymus vulgaris* EO (0.2% *v/v*) | | 86% | | | |
| | | *Melaleuca alternifolia* EO (2% *v/v*) | | 71.5% | | | |
| | | *S. aromaticum*-based formulation (0.4% *v/v*) + pinolene | Seed coating | From 6% to 80% | Rate of seed germination equal to the control | | |
| | | *T. vulgaris* EO (0.3% *v/v*) + pinolene | | 53% | | | |
| | | *M. alternifolia* EO (2% *v/v*) + pinolene | | 5% | | | |
| Maize (*Zea mays*) | *F. verticillioides* [2] | *Jacaranda mimosifolia* WE (0.6% *v/v*) | Immersion for 1 h | Pot experiment: 75% Field experiment: 64% | – | Induction of defense-related enzymes | [52] |
| | *F. equiseti* [2]; *F. culmorum* [2]; *F. poae* [2]; *F. avenaceum* [2] | *S. aromaticum* EO ($5 \times 10^4$ ppm) | Immersion for 8 min | | Total inhibition of seed germination | Inhibition of pathogen development | [46] |
| | *Aspergillus* spp. [2] | Ozone (60 mg L$^{-1}$) | Ozonation for 480 min | 99.7% | – | Aflatoxins and microbial contamination reduction | [53] |
| | *Fusarium* spp. [2] | | | 99.9% | | | |
| | *Aspergillus* spp. [1] | Ozone (2.14 mg L$^{-1}$) | Ozonation for 50 h | 78.5% | – | Pathogen incidence reduction | [54] |
| | *Penicillium* spp. [1] | | | 98.0% | | | |
| Tomato (*Solanum lycopersicum*) | Fusarium wilt *F. oxysporum* * | *Artemisia absinthium* EO (0.5 mg mL$^{-1}$) | Seed coating | Reduction in disease symptoms. | Rate of seed germination equal to the control | Induction of a long-term response (ROS production and callose deposition) | [55] |
| | | *Eucalyptus grandis* EO (6% *v/v*) | Immersion | 73.0% | Rate of seed germination equal to the control | | [48] |
| | | *Cuminum cyminum* EO (6% *v/v*) | | 53.1% | | | |
| | | *Citrus sinensis* EO (6% *v/v*) | | 84.3% | | | |
| | *F. oxysporum* f. sp. *lycopersici* * | *Origanum vulgare* EO 1200 µg mL$^{-1}$ | Immersion | 52.0% | No phytotoxicity | Reduction in percentage disease severity and incidence | [49] |
| Squash (*Cucurbita maxima*) | *Stagonosporopsis cucurbitacearum* [1] and seven other fungal species | *Cymbopogon citratus* EO and six other essential oils. (0.5 mg mL$^{-1}$) | Immersion for 6 h | From 67% to 84.4% | | Seedling emergence increasing | [56] |
| Bean (*Phaseolus vulgaris*) | Anthracnose/ *Colletotrichum lindemuthianum* [2] | *Ocimum gratissimum* EO (80 mg kg$^{-1}$) | Immersion | Anthracnose symptoms reduction of 73.9% | Rate of seed germination equal to the control | | [57] |
| | | *S. aromaticum* EO (80 mg kg$^{-1}$) | | Anthracnose symptoms reduction of 65.5% | | | |

**Table 3.** *Cont.*

| Crop | Target Disease/Pathogen | Substance (Concentration) | Application | Effectiveness (Disease/Symptoms Reduction) | Possible Phytotoxicity | Activity/Defense Response | Reference |
|---|---|---|---|---|---|---|---|
| **Potential Basic Substances—Fungi and Oomycetes** | | | | | | | |
| Lettuce (*Lactuca sativa*) | *Cladosporium* sp. [1] | *Eugenia caryophyllus* EO (500 µL L$^{-1}$) | | 86.0% | Seed germination reduction | | [58] |
| | *Alternaria* sp. [1] | | | 70.0% | | | |
| | *Cladosporium* sp. [1] | *Cymbopogon citratus* EO (500 µL L$^{-1}$) | | 98.0% | | | |
| | *Alternaria* sp. [1] | | | 85.0% | | | |
| | *Cladosporium* sp. [1] | *Rosmarinus officinalis* EO (500 µL L$^{-1}$) | | 33.0% | | | |
| | *Alternaria* sp. [1] | | | 7.5% | | | |
| Onion (*Allium cepa*) | *A. alternata* [1] | *Abies alba* EO (0.2 µL cm$^{-3}$) | Immersion for 6 h | 10.4% | Rate of seed germination equal to the control | | [59] |
| | *Botrytis allii* [1] | | | 80.5% | | | |
| | *B. cinerea* [1] | | | 76.9% | | | |
| | *Cladosporium* spp. [1] | | | 28.5% | | | |
| | *Fusarium* spp. [1] | | | 84.2% | | | |
| | *A. alternata* [1] | *Pinus sylvestris* EO (0.2 µL cm$^{-3}$) | Immersion for 6 h | 16.3% | | | |
| | *Botrytis allii* [1] | | | 55.5% | | | |
| | *B. cinerea* [1] | | | 88.4% | | | |
| | *Cladosporium* spp. [1] | | | 7.1% | | | |
| | *Fusarium* spp. [1] | | | 84.2% | | | |
| | *A. alternata* [1] | *T. vulgaris* EO (0.2 µL cm$^{-3}$) | Immersion for 6 h | 10.4% | | | |
| | *Botrytis allii* [1] | | | 80.5% | | | |
| | *B. cinerea* [1] | | | 100% | | | |
| | *Cladosporium* spp. [1] | | | 35.7% | | | |
| | *Fusarium* spp. [1] | | | 94.7% | | | |
| Sunflower (*Helianthus annuus*) | *Plasmopara halstedii* [1] | *Nigella sativa* EO (0.6%) | Spray | Decrease in sporangium quantity 70.1% | – | | [60] |
| | | *Sambucus nigra* EO (0.6%) | | 87.3% | | | |
| | | *Hypericum perforatum* EO (0.6%) | | 90.5% | | | |
| | | *Allium sativum* EO (0.6%) | | 90.0% | | | |
| | | *Vitis vinifera* EO (0.6%) | | 91.2% | | | |
| | | *Zingiber officinale* EO (0.6%) | | 90.2% | | | |

[1] Natural contamination; [2] artificial inoculation; * soil contamination; EO = essential oil; WE = water extract.

Naturally contaminated *Colletotrichum lindemuthianum* beans were treated with basil (*Ocimum gratissimum*) and clove EOs, and the treatment caused a significant reduction in anthracnose incidence without affecting the germination and the emergence speed index [57]. Lemongrass (*Cymbopogon citratus*), lavender (*Lavandula dentata*), lavandin (*Lavandula hybrida*), tea tree (*Melaleuca alternifolia*), bay laurel (*Laurus nobilis*), and two different marjoram (*Origanum majorana*) EOs were tested as seed treatments against the main *Cucurbita maxima* seedborne fungal pathogens: *Stagonosporiopsis cucurbitacearum*, *Alternaria alternata,* and *F. solani* [56]. The seed immersion treatments were carried out at a concentration of 0.5 mg mL$^{-1}$ for 6 h, with mixing every 30 min, and the results showed that the incidence of multiple seedborne fungal pathogens was significantly reduced on squash seeds, with no negative effect on germination. In addition, the *C. citratus* EO increased seedling emergence and reduced the incidence of *S. cucurbitacearum* in plantlets.

Waureck et al. [58] found that the main fungi observed in organic and untreated lettuce seeds were *Cladosporium* sp. and *Alternaria* sp. seed, and treatments with clove, lemongrass,

and rosemary EOs at a dose of 0.5% (*v/v*) significantly reduced their presence on seeds, but with negative effects on germination, suggesting that the application dose of these essential oils should be modulated for lettuce seeds [58].

Exogenous application of specific plant extracts can induce resistance in the host plant via higher levels of host defense enzymes and PR protein stimulation. An absinthium (*Artemisia absinthium*) EO seed coating was tested on tomato seeds and was able to protect seed germination and seedling growth, priming tolerance in tomato seedlings previously infected with *F. oxysporum* f.sp. *lycopersici* by the induction of metabolic changes responsible for the long-term tolerance of the tomato [55]. An extract of *Jacaranda mimosifolia* (1.2%) applied to maize seeds provided significant protective effects on plants compared to the inoculated control, by also inducing a systemic resistance in the host plants [52].

Silver fir (*Abies alba*), pine (*Pinus sylvestris*), and thyme EOs were tested as seed treatments on onion by immersion for 6 h, and seed health test on potato dextrose agar showed that all the oil treatments effectively controlled *Fusarium* spp. on the onion seeds and frequently reduced their infestation with *Botrytis* spp. The lowest dose tested with antifungal activity and without phytotoxic effects was $0.2\ \mu L\ cm^{-3}$, while increasing the dose led to increased phytotoxicity [59].

Commercial EOs obtained from different parts of black cumin (*Nigella sativa*), mustard (*Sambucus nigra*), St. John's wort (*Hypericum perforatum*), garlic (*Allium sativum*), grape (*Vitis vinifera*), and ginger (*Zingiber officinale*) plants were evaluated in vivo against the oomycete *Plasmopara halstedii*. The application of the above oils as a spray seed treatment was shown to provide protection against mildew in sunflower plants under in vivo conditions, assessed as a percentage reduction in the sporangium count ranging from 70.1% to 90.5% [60].

In order to obtain the best advantages from the volatile nature of active compounds, oregano, thyme (*Thymus vulgaris*), and coriander (*Coriandrum sativum*) EOs were tested in vapor form for their antifungal potential against *Alternaria* spp., *Fusarium* spp. and *Drechslera* spp. infection on wheat seeds [50]. Wheat seeds were stored in an atmosphere enriched with essential oil vapors and a selective antifungal effect was highlighted as the following: oregano EO and thyme EO significantly inhibited *Alternaria, Fusarium*, and *Drechslera* (that was the most sensitive). Regarding the phytotoxic effects of EO vapors on the germination of the seeds, thyme EO and oregano EO had an inhibitory effect, especially at 0.4%. This effect was cumulative over time. The EOs inhibited deoxynivalenol (DON) occurrence, and the maximum percentage of inhibition was obtained after 21 days of vapor exposure, with the most effective timing being when applied at 0.2%.

Ozone has been declared as a generally recognized as safe (GRAS) substance and its application in agriculture has increased in recent years (Table 2) [61]. Ozone gas was applied on maize and wheat seeds for fungal decontamination: ozone gas application for 300 min at a rate of $60\ mg\ L^{-1}$ was able to reduce the incidence of *Aspergillus* spp. and *Penicillium* spp. (both ~ 54%) on artificially infected wheat seeds [51], while 50 h application at a rate of $2.14\ mg\ L^{-1}$ reduced *Aspergillus* spp. (78.5%) and *Penicillium* spp. (98.0%) incidence on naturally infected maize seeds [54]. Thanks to its oxidizing properties, ozonation can also represent an effective method for the remediation of cereals contaminated by mycotoxins, where gaseous ozone application for 480 min at the rate of $60\ mg\ L^{-1}$ reduced aflatoxins and microbial contamination in corn artificially infected with *Aspergillus* spp. and *Penicillium* spp. [53].

### 4.2. Activity of Potential Basic Substances against Bacteria

Several studies have investigated the effects of potential basic substances to control bacterial seedborne pathogens (Table 4). Kotan et al. [62] revealed the antibacterial effects of different extracts of *Origanum onites* (hexane, acetone, and chloroform) on tomato and lettuce seeds inoculated with *Clavibacter michiganensis* ssp. *michiganensis*, *Xanthomonas axonopodies* pv. *vesicatoria*, and *X. campestris* pv. *zinniae*. Extracts were applied by seed soaking after inoculation. The hexane extract was the most effective against *C. michiganensis* ssp. *michiganensis*, with a 75% disease severity reduction at $15\ mg\ mL^{-1}$, whereas the

chloroform extract was more effective against *X. axonopodies* pv. *vesicatoria* and *X. campestris* pv. *vitians*, with a 77% reduction at 20 mg ml$^{-1}$ and a 74% reduction at 15 mg mL$^{-1}$, respectively. The authors attributed this strong antibacterial activity to the presence of carvacrol and thymol, two of EO's major constituents. No phytotoxicity was found on seeds treated with all the extracts tested; indeed, different extracts even increased seed germination and plant height in tomato seedlings at concentrations of 5 and 10 mg mL$^{-1}$. A study by Karabüyük and Aysan [63] on the reduction in bacterial speck disease caused by *Pseudomonas syringae* pv. *tomato* demonstrated that immersion treatments of tomato seeds with aqueous extracts of *Zingiber officinale* and *Origanum vulgare* (Istanbul thyme) reduced 100% of bacterial speck disease incidence and severity on tomato seedlings. In addition, aqueous extracts of *Eucalyptus camaldulensis* and *Allium sativum* reduced disease incidence and severity by 98%–97% and 99.3%–56.8%, respectively, whereas coriander extracts only reduced disease incidence by up to 63%. All the tested extracts did not affect seed germination. The antimicrobial activity of thyme EO on soybean seeds infected with *P. savastanoi* pv. *glycinea* B076 and *P. syringae* M7-C1, causal agents of bacterial blight in soybean, was investigated at a greenhouse scale by Sotelo et al. [64]. The results obtained demonstrated that 1.76 mg mL$^{-1}$ of the essential oil previously diluted in skim milk powder reduced the number of phytopathogenic bacteria inoculated on the seeds by about 6 logs. In addition, the germination of the treated seeds was 73%, whereas for the infected seeds it was near 50%. Similarly, the disease incidence of soybean plants from infected seeds and treated with thyme EO was reduced by 24.05% for *P. syringae* M7-C1 and by 29.76% for *P. savastanoi* pv. *glycinea* B076. Another study [65] focused on the plant pathogenic bacteria *Burkholderia glumae*, a rice seedborne pathogen that causes grain rot in rice plants, and showed that immersion treatment of rice seeds for 10 min with clove EO at 2% and 5% *v*/*v* and citronella (*Cymbopogon nardus*) EO at 1% and 3% *v*/*v* reduced by 50% the disease incidence in plants, with the 5% clove oil treatment giving the highest rice grain production. However, no phytotoxicity data were provided. *Cistus ladaniferus* subsp. *ladanifer* EO, together with its methanolic and ethanolic extracts, and *Mentha suaveolens* EO, were used for the treatment of tomato seeds infected with the phytopathogenic bacterium *C. michiganensis* subsp. *michiganensis* [66]. The results evidenced that C. *ladaniferus* subsp. *ladanifer* oil and extracts and *Mentha suaveolens* EO inhibited in vitro the growth of *C. michiganensis* with a minimal inhibitory concentration (MIC) of 0.78 mg mL$^{-1}$, but the in vivo treatment with such EOs at MIC and 4 × MIC showed a negative effect on tomato seed germination. On the contrary, treatment with ethanolic and methanolic extracts of *C. ladaniferus* showed no phytotoxicity, with the methanolic extract revealing the highest percentages of germination. Treatments were performed by soaking the seeds for 1 h. In another study on the tomato, two other EOs (cinnamon and oregano) were tested in vivo for their antibacterial activity against *C. michiganensis* subsp. *michiganensis* [67]. Artificially infected tomato seeds were treated by immersion with these two oils at a concentration of 0.4% and their efficacy in controlling the pathogen was evaluated using a real-time PCR molecular assay for in planta bacterial quantification at the very first stage of development: both oils significantly reduced the bacterial presence in seedlings compared to controls (untreated and water-treated), with oregano being the most effective. Oregano EO showed no phytotoxicity at the concentrations tested up to 0.4%, while cinnamon EO had little effect on germination, reducing it by one or two percentage points.

**Table 4.** In vivo and in-field activities of potential basic substances applied as seed treatments to control seedborne bacteria in different crops. Effectiveness is reported as the disease incidence or symptom percentage reduction compared to the untreated control. Phytotoxicity was evaluated through germination testing and the results are compared to the untreated control.

| | | | | Potential Basic Substances—Bacteria | | | |
|---|---|---|---|---|---|---|---|
| Crop | Target Disease/Pathogen | Substance (Concentration) | Application | Effectiveness (Disease/Symptoms Reduction) | Possible Phytotoxicity | Activity/Defense Response | Reference |
| Tomato (*Solanum lycopersicum*) | *Clavibacter michiganensis* subsp. *michiganensis* [1] | *Cinnamomum zeylanicum* EO (0.4% *v/v*) | Immersion | 25% | Germination reduced by 1–2% | Bactericidal activity | [67] |
| | | *Origanum vulgare* EO 0.4% (*v/v*) | Immersion | 100% | Rate of seed germination equal to the control | | |
| | *C. michiganensis* subsp. *michiganensis* [1] | *O. onites* HE (15 mg mL$^{-1}$) | Immersion | 75% | Rate of seed germination equal to the control | Different extracts increased seed germination and plant height | [62] |
| | *Xanthomonas axonopodies* pv. *vesicatoria* [1] | *O. onites* CE (20 mg mL$^{-1}$) | | 76.91% | | | |
| | *X. campestris* pv. *zinniae* [1] | *O. onites* chloroform extract (15 mg mL$^{-1}$) | | 74.22% | | | |
| | *Pseudomonas syringae* pv. *tomato* [1] (Pst) | *Zingiber officinale* AE | Immersion | 100% | Rate of seed germination equal to the control | | [63] |
| | | *O. vulgare* L. AE (Istanbul thyme and Izmir thyme) | | 100% | | | |
| | | *Eucalyptus camaldulensis* AE | | 98% (incidence) 97% (severity) | | | |
| | | *Allium sativum* AE | | 99% (incidence) 57% (severity) | | | |
| | | *Coriandrum sativum* extracts | | Up to 63% (incidence) | | | |
| Soybean (*Glycine max*) | *P. savastanoi* pv. *glycinea* B076 [1] | *Thymus vulgaris* EO (1.76 mg mL$^{-1}$) | | 24.05% | Seed germination increasing | Increasing seed germination | [64] |
| | *P. syringae* M7-C1 [1] | | | 29.76% | | | |
| Rice (*Oryza sativa*) | *Burkholderia glumae* [1] | *S. aromaticum* EO *Cymbopogon nardus* | | 50% | Rate of seed germination equal to the control | | [65] |
| Tomato (*Solanum lycopersicum*) | *C. michiganensis* subsp. *michiganensis* [1] | *Cistus ladaniferus* subsp. *ladanifer* EO | Immersion for 1 h | Minimal inhibitory concentration (MIC): 0.78 mg mL$^{-1}$ | Rate of seed germination equal to the control | Bacterial growth inhibition | [66] |
| | | *Cistus ladaniferus* subsp. *ladanifer* ME | | | Seed germination increasing | | |
| | | *Mentha suaveolens* EO | | | Rate of seed germination equal to the control | | |

[1] Natural contamination; EO = essential oil; AE = aqueous extract; ME = methanolic extract; CE = chloroform extract; HE = hexane extract.

### 4.3. Activity of Potential Basic Substances against Viruses and Phytoplasma

Basic substances or potential basic substances having a direct action on viruses or phytoplasma inside plant cells are nowadays quite unknown. Research directly targeting these pathogens inside the plant host cells by applying sustainable means of control is useful and highly recommended. Stommel and colleagues demonstrated [68] that exposure of pepper mild mottle virus to ozone resulted in viral inactivation, but at insufficient levels to prevent viral transmission from highly contaminated pepper (*Capsicum annuum*) seeds (Table 5). Viruses and phytoplasma are non-culturable organisms; therefore, it is not easy

to verify their direct effects on pathogens and just in vivo trials can be used. However, in vivo trials are much more complex and require infected materials with a high load of the pathogen to gain significant results. Virus and phytoplasma can be controlled by physical treatments such as thermotherapy or by controlling their insect vectors. Basic substances or potentially basic substances can also effectively be used against the vectors to reduce the spread of viruses and phytoplasma.

**Table 5.** In vivo and in-field activities of potential basic substances applied as seed treatments to control seedborne viruses in different crops. Effectiveness is reported as the disease incidence or symptom percentage reduction compared to the untreated control. Phytotoxicity was evaluated through germination testing and results are compared to the untreated control.

| Potential Basic Substances—Viruses | | | | | | | |
|---|---|---|---|---|---|---|---|
| Crop | Target Disease/ Pathogen | Substance (Concentration) | Application | Effectiveness (Disease/Symptoms Reduction) | Possible Phytotoxicity | Activity/Defense Response | Reference |
| Pepper (*Capsicum annum*) | Pepper mild mottle virus (PMoV) [1] | Ozone (20 ppm) | Ozonation for 14 h | Inactivation of the seedborne virus; however, at high seed contamination levels, this treatment was insufficient to prevent infection | Rate of seed germination equal to the control | | [68] |

[1] Natural contamination.

## 5. Conclusions

Sowing high-quality seeds is important to reduce yield losses. Seed treatment is an essential step in the management of crops diseases. This step can play economic and environmental roles in reducing the cost and quantity of pesticides in the field. Sustainable seed treatments using basic substances and potential basic substances can be good alternatives to controlling the main seedborne pathogens and for promoting more sustainable crop systems. Basic substances have a registration cost that is much lower than the one of synthetic pesticides (EUR 50,000 versus EUR 300 million) [15], and small companies can also promote the application of a basic substance. There is relatively poor information about the effectiveness of basic substances and potential basic substances as seed treatments compared to synthetic pesticides, but in some research, their effectiveness can be considered comparable or slightly lower than the one of synthetic fungicide. The cost of the product is comparable to or slightly higher than synthetic pesticides. Conversely, by applying basic substances, there are no issues with the safety of the treated commodities and a there is a lower impact on the environment. These alternatives now need to be further developed as appropriate seed treatments for ensuring global food security in a green way.

**Author Contributions:** Conceptualization, L.O. and G.R.; investigation, L.O., M.B.A., C.C.-L., M.M. (Marwa Moumni), M.M. (Monica Mezzalama) and J.B.M.-H.; writing—original draft preparation, L.O. and M.M. (Marwa Moumni); writing—review and editing, L.O., M.B.A., C.C.-L., M.M. (Marwa Moumni), M.M. (Monica Mezzalama) and J.B.M.-H.; supervision, G.R. and L.O.; funding acquisition, G.R. and L.O. All authors have read and agreed to the published version of the manuscript.

**Funding:** This work was carried out within the "Euphresco Basic substances as an environmentally friendly alternative to synthetic pesticides for plant protection (BasicS)" project (Objective 2020-C-353). G.R. acknowledges the support of the PSR Marche Project "CleanSeed".

**Data Availability Statement:** All data are contained in the article.

**Acknowledgments:** This review is dedicated to Luca Riccioni, who greatly supported the researchers and the project activities and recently suddenly passed away prematurely. Many thanks are also expressed to Baldissera Giovani, the Euphresco Coordinator, who promoted and supported the project Euphresco BasicS along with its development.

**Conflicts of Interest:** The authors declare no conflict of interest.

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
