# Peer review of "Basic Substances and Potential Basic Substances: Key Compounds for a Sustainable Management of Seedborne Pathogens"

_horticulturae, doi:10.3390/horticulturae9111220_

Round 1

Reviewer 1 Report

Comments and Suggestions for Authors

In this manuscript, the authors summarized the latest advances in the application methods and effect of basic substances and potential basic substances on the management of seedborne pathogens. This review is informative and instructive for the researchers of this area. The manuscript is well-written, and could be accepted after a minor revision. Here are some minor issues need to be addressed:

1 The readers may also concern about the efficiency and cost of the basic substances and potential basic substances on the management of seedborne pathogens.

2 The tables need to be reorganized. For example, the words should not be separated into different lines. Besides, Table 1 at line 281 should be Table 3. 0,5% v/v in Table 1 should be revised.

3 The writing of units should be uniformed. For example, the mg L-1 was used at line 345, while mg/ml was used at line 362. 43%-78% at line 211 and 4.7-26.9% in table 1 should be corrected.

4 Both straight and italic font of ‘In vivo’ were used. Please make sure the ‘x’ in 5x103 and 5x104 at line 269 was correctly typed.

Author Response

Thanks for your comments.

1) We added the following sentences.

  • Lines 454-457: "Basic substances have a registration cost that is much lower than the one of synthetic pesticides (50 000 Euro versus 300 million Euro) [15], then also small companies can promote the application of a basic substance."
  • Lines 459-463: "Basic substances have a registration cost that is much lower than the one of synthetic pesticides (50 000 Euro versus 300 million Euro) [15], then also small companies can promote the application of a basic substance."

2) We have reorganized the tables, modified the format and style to make them easier to read and we revised what was addressed.

3) The writing of units were corrected in all the manuscript (text and tables)

4) The font for ‘in vivo’ was corrected.

Reviewer 2 Report

Comments and Suggestions for Authors

The manuscript ID: horticulturae-2651446, “BASIC SUBSTANCES AND POTENTIAL BASIC SUBSTANCES: KEY COMPOUNDS FOR A SUSTAINABLE MANAGEMENT OF SEEDBORNE PATHOGENS has been reviewed.
The authors in the manuscript (review) BASIC SUBSTANCES AND POTENTIAL BASIC SUBSTANCES: KEY COMPOUNDS FOR A SUSTAINABLE MANAGEMENT OF SEEDBORNE PATHOGENS” showed a significant problem regarding the provide an overview of the use of already approved basic substances and of potential basic substances as seed treatments for the control of seedborne pathogens and techniques that may be used for the best application method. In This review, the authors used substances approved by the EU: chitosan hydrochloride, chitosan, vinegar, mustard seed powder and hydrogen peroxide, as well as potential basic substances such as ozone, essential oils, plant extracts, which were used in numerous studies to treat seeds.The authors described methods for seed treatment with consideration: seed immersion and seed dressing and coating oraz wskazali na ich wykorzystanie i skutki działania. They described seed treatment various plant species with approved basic substances against fungi and oomycetes: chitosan, mustard seed power, vinegar and hydrogen  peroxide what they showed in table 1. However, in my opinion, this table is not very readable and requires correction (in some places the descriptions from individual columns are combined). They also showed activity of approved basic substances (hydrogen peroxide) against bacteria. They described the possibilities of use seed treatment with potential basic substances (essential oils, plant extract, ozon) against pathogens (fungi and oomycetes, bacteria, viruses and phytoplasma). In my opinion, tables 3 and 4  are not very readable and requires correction. On page 9, Table 1 should read Table 3. On page 16, line 355 (Table 3) there should be (Table 4). I also suggest that authors provide an extended description of its use of potential basic substances against viruses and phytoplasma.

This is a good manuscript that uses good references sources. With minor revisions, I recommend it for publication in Horticulturae (MDPI).

Author Response

Thank you for your comments. We have reorganized the tables 1, 2, 3 and 4, modifying the format and style to make them easier to read and we revised what was addressed. Concerning your suggestion to provide an extended description of the use of potential basic substances against viruses and phytoplasma, we added the following sentences in lines 439-442:

"Viruses and phytoplasma are non culturable organisms, therefore it is not easy to verify the direct effects on the pathogen and just in vivo trials can be used. However, in vivo trials are much more complex and require infected materials with high load of pathogen to gain significant results."

Reviewer 3 Report

Comments and Suggestions for Authors

This paper is reviewing recent information on application of basic substances and potential basic substances for seed treatment. Literature sources are sound and core contents were given in the text. The review covered twenty-four basic substances for seed treatment which are currently approved in the EU, basic substances under evaluation and may be approved soon for seed Overall, Information is useful allow us to understand the status of EU regulation on the seed treatment substances. Regarding manuscript format, the style of Tables needs to be little changed to be more readable. Current form is low in readability.

Author Response

Thank you for your comments. We modified the format and the style to make it more readable.